# Response to comment on 'The clinical pharmacology of tafenoquine in the radical cure of *Plasmodium vivax* malaria: An individual patient data meta-analysis'

**James A Watson[1,2,3]\*, Robert J Commons[3,4], Joel Tarning[2,5], Julie A Simpson[6], Alejandro Llanos Cuentas[7], Marcus VG Lacerda[8], Justin A Green[9†], Gavin CKW Koh[10], Cindy S Chu[2,11], François H Nosten[2,11], Richard N Price[2,3,4], Nicholas PJ Day[2,5], Nicholas J White[2,5]\***

[1]Oxford University Clinical Research Unit, Hospital for Tropical Diseases, Ho Chi Minh, Viet Nam; [2]Centre for Tropical Medicine and Global Health, Nuffield Department of Medicine, University of Oxford, Oxford, United Kingdom; [3]WorldWide Antimalarial Resistance Network, Oxford, United Kingdom; [4]Global Health Division, Menzies School of Health Research, Charles Darwin University, Darwin, Australia; [5]Mahidol Oxford Tropical Medicine Research Unit, Faculty of Tropical Medicine, Mahidol University, Bangkok, Thailand; [6]Centre for Epidemiology and Biostatistics, Melbourne School of Population and Global Health, University of Melbourne, Melbourne, Australia; [7]Unit of Leishmaniasis and Malaria, Instituto de Medicina Tropical "Alexander von Humboldt", Universidad Peruana Cayetano Heredia, San Martín de Porres, Peru; [8]Fundação de Medicina Tropical Dr Heitor Vieira Dourado, Manaus, Brazil; [9]GSK, Brentford, United Kingdom; [10]Department of Infectious Diseases, Northwick Park Hospital, Harrow, United Kingdom; [11]Shoklo Malaria Research Unit, Mae Sot, Thailand

**\*For correspondence:**
james@tropmedres.ac (JAW);
nickw@tropmedres.ac (NJW)

**Present address:** [†]AstraZeneca, Cambridge, United Kingdom

**Abstract** In our recent paper on the clinical pharmacology of tafenoquine (Watson et al., 2022), we used all available individual patient pharmacometric data from the tafenoquine pre-registration clinical efficacy trials to characterise the determinants of anti-relapse efficacy in tropical vivax malaria. We concluded that the currently recommended dose of tafenoquine (300 mg in adults, average dose of 5 mg/kg) is insufficient for cure in all adults, and a 50% increase to 450 mg (7.5 mg/kg) would halve the risk of vivax recurrence by four months. We recommended that clinical trials of higher doses should be carried out to assess their safety and tolerability. Sharma and colleagues at the pharmaceutical company GSK defend the currently recommended adult dose of 300 mg as the optimum balance between radical curative efficacy and haemolytic toxicity (Sharma et al., 2024). We contend that the relative haemolytic risks of the 300 mg and 450 mg doses have not been sufficiently well characterised to justify this opinion. In contrast, we provided evidence that the currently recommended 300 mg dose results in sub-maximal efficacy, and that prospective clinical trials of higher doses are warranted to assess their risks and benefits.

## Introduction

Most antimalarial drugs have required dose optimisation following their initial introduction. It appears likely that tafenoquine is not an exception. The optimal doses of the 8-aminoquinoline drugs

(primaquine and tafenoquine) in the prevention of *Plasmodium vivax* relapse are a trade-off between the blood stage and hypnozoite effects (benefiting all patients), and the haemolytic risk in the sub-group of patients with glucose-6-phosphate dehydrogenase (G6PD) deficiency. The dose-response relationship for tafenoquine-induced haemolysis in G6PD deficiency has not been well characterised (the entirety of the available data for the 300 mg dose comprises 3 heterozygous females with >40% enzyme activity, *Rueangweerayut et al., 2017*) but the risk is regarded as substantial, which is why the drug is restricted to patients with G6PD activity >70%. Compared with primaquine, the most widely used 8-aminoquinoline antimalarial, tafenoquine, if not used appropriately, may have a higher risk of causing severe haemolysis. Primaquine can be stopped once haemolysis is apparent, and the effect is short-lived. Tafenoquine has an elimination half-life of approximately 16 days and cannot be "stopped". As high doses (up to 15 mg/kg) of tafenoquine are well tolerated in G6PD normal individuals (the substantial majority of patients, *Watson et al., 2022*), and prevention of relapse provides substantial health benefits (*Dini et al., 2020*), our individual patient data meta-analysis focused on efficacy. We concluded that the current dose has sub-optimal efficacy, and recommended evaluation of higher doses.

Sharma et al challenge our findings. They "assert that, collectively, these data confirm that the benefit–risk profile of a single 300 mg dose of tafenoquine [...] continues to be favourable" (*Sharma et al., 2024*). We believe that this opinion is not supported by the facts. There is no need to wait for 'real-world evidence' to confirm superior efficacy of higher tafenoquine doses when the available clinical trial data demonstrate this relationship beyond reasonable doubt. Instead, randomised clinical trials of higher tafenoquine doses are needed to characterise their efficacy, safety and tolerability.

The following provides a point-by-point reply to Sharma et al.

Sharma et al. state that in the DETECTIVE study 'doubling the tafenoquine dose from 300 mg to 600 mg was associated with only a marginal increase (from 89.2% to 91.9%) in the primary efficacy endpoint'.

The DETECTIVE phase 2 trial was small. Only 57 patients received 300 mg and 56 received 600 mg, distributed across four different countries. One of the countries was India where long latency relapse strains are found. These relapses would have occurred after 6 months (the study's follow up duration). There was substantial variation in body weight. Direct comparison between the two doses ignores body weight variation (i.e. the major determinant of efficacy) and is underpowered. We fit a logistic regression model to the DETECTIVE phase 2 efficacy data. We estimate an odds ratio for any recurrence at 6 months of 0.67 per mg/kg increase (95% CI 0.58–0.76). At 4 months this odds ratio is 0.62 (95% CI 0.51–0.72), i.e. almost identical to our results from the pooled dataset (*Watson et al., 2022*).

Sharma et al. ask what we mean by 'optimal primaquine regimens', and whether these are 'WHO-recommended schedules of primaquine or regimens defined as optimal based on nonregulatory studies of primaquine'. We note that there have been no regulatory studies of primaquine. A primaquine total dose of 7 mg/kg is approved by WHO for the Southeast Asian region. A pooled individual patient data meta-analysis shows that this dose is clearly more efficacious than 3.5 mg/kg (*Commons et al., 2023*).

## Results
### Efficacy models employed by Watson et al

Sharma et al. state that 'Details of how the best predictor was selected and how statistical significance was judged were not provided'. The code for the statistical analysis is openly accessible. For the main analysis it is provided as an RMarkdown file (GitHub, copy archived at *Watson, 2023*). As expected, the plasma tafenoquine AUC and $C_{max}$ values are highly correlated with the mg/kg dose, whereas the terminal elimination half-life is not. For the AUC and the $C_{max}$, we compared results for analyses which did and did not adjust for the mg/kg dose. When adjusting for the mg/kg dose, there was no longer a clear relationship between AUC and the odds of recurrence (95% CI for the adjusted odds ratio is 0.64–1.23), or the $C_{max}$ and recurrence (95% CI for the adjusted odds ratio is 0.69–1.49). The corresponding code can be found on lines 981 and after (section 'AUC'), and lines 1023 and after (section 'CMAX') in the main RMarkdown file.

> ## Box 1. Why a CART model using the Gini criterion is an inappropriate choice for the estimation of the optimal tafenoquine dose.
>
> Suppose $X \sim Normal(0, 1)$ and $Y \sim Bernoulli(p)$ where $p = e^x/(1 + e^x)$. Then, for large positive value of $x$, $E[y|x] \approx 1$ and for large negative value of $x$, $E[y|x] \approx 0$. Given a large sample realisation of the random variables $X$ and $Y$ under this data generating process, the optimal split for the Gini criterion would be $x = 0$ (***Figure 1***). If $y = 1$ was 'no recurrence' and $y = 0$ 'recurrence', it is unclear why the optimal value of a determinant x should be the value for which the probability of recurrence is 0.5. 'Clinical relevance' implies a value proposition, whereby a utility (or loss) is assigned to each outcome. In terms of efficacy and tolerability only, an optimal dose would be in an area where the dose-response is flat (i.e where increases in the dose lead to very marginal increases in efficacy), not where it is the steepest.

## Use of a 4-month versus 6-month follow-up period

The optimal duration of follow-up in a study of radical curative efficacy remains debated. In most tropical regions *P. vivax* relapses are highly predictable and occur within a few weeks to months after initial treatment. Trials which have included no 8-aminoquinoline treatment arms indicate that >90% of first relapses occur within four months (***Commons et al., 2023***). Longer follow-up increases sensitivity (more relapses included) but lowers specificity (more reinfections included). The results of our analysis are almost identical when a 6 month endpoint was applied. This was shown in Appendix 1—figures 4 and 5 in ***Watson et al., 2022***.

Sharma et al. consider that we provide insufficient detail of the sensitivity analysis which included patients receiving 300 mg only. The sensitivity analysis is given on line 501 in the section "Logistic Regression" in the main `RMarkdown` (`TQ_efficacy.Rmd`). The exact same analysis as the primary analysis was performed on the restricted subset of patients who received a 300 mg dose of tafenoquine.

Sharma et al. ask how the dosing bands were chosen for our Figure 2. The selected bands were chosen for simplicity of visualisation ensuring enough patients fell into each category and were thus meaningful. Importantly, no quantitative results depend upon this categorisation.

## Rationale for tafenoquine dose selection

Sharma et al. state that we did not discuss 'the classification and regression tree analysis, in which a clinically relevant breakpoint tafenoquine AUC value of 56.4 µg·h/mL was identified'.

The relevance of the quoted 56.4 µg.h/ml threshold for AUC is unclear. We do not see why this threshold as determined should be considered "clinically relevant". A CART model determines optimal breakpoints by minimising a given loss function. It appears from the cited publication (***Tenero et al., 2015***) that the authors used the default parameters in the rpart function in R (from the rpart package). If this is correct, then it would imply that the CART model was fit using the Gini criterion, thus maximising the 'purity' of the split (recurrence vs no recurrence). The Gini criterion is not an appropriate choice. Under the Gini criterion the cut-off threshold is the value which equally balances sensitivity and specificity for a continuous covariate which has a continuous relationship with the outcome (e.g. a linear relationship on the log-odds scale). ***Box 1*** gives a hypothetical example (where $X$ is analogous to weight) whereby the Gini criterion selects as optimal the value for which the probability of recurrence is exactly 0.5. We do not believe that this should be considered optimal.

Sharma et al. quote the TEACH study (paediatric tafenoquine study) as validation of the AUC approach to tafenoquine dose selection (***Vélez et al., 2022***). Firstly, we note that TEACH was a single arm, non-randomised study with no control group. Hence "efficacy" (a causal concept) is ill-defined. Secondly, we note that the mg/kg doses in TEACH were higher on average than in the adult efficacy studies. Using the reported mean body weights (Table 1 in ***Vélez et al., 2022***), the mean dose was approximately 7.0 mg/kg.

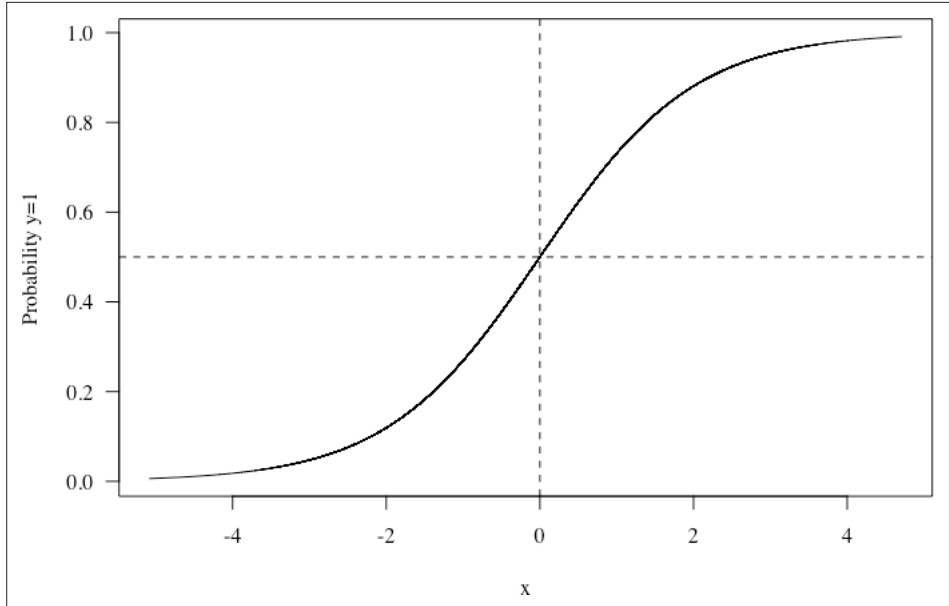

**Figure 1.** Hypothetical example showing why the Gini criterion does not lead to a split which is clinically relevant in this context. Under this data generating process, the optimal split for the Gini criterion is at x=0, corresponding to a probability of 0.5 that y=1.

Sharma et al. state that in the INSPECTOR study (*Sutanto et al., 2023*), body weight was not a significant predictor of efficacy. Firstly, no coefficient values or confidence intervals were provided in the cited publication and a p-value >0.05 does not prove the null hypothesis. Secondly, the tabular results presented in the paper suggest that the efficacy of tafenoquine was greater in the lighter patients. The failure rate was 67% in the patients weighing ≤69kg, but was 91% in the patients weighing >69 kg (odds ratio for failure is 5.3 [95%CI 1.0–27.5; *P*=0.07]).

## Use of an 'unvalidated' biomarker

Sharma et al. dispute our causal model for tafenoquine because 'Central to this model are methaemoglobin (MetHb) production and active metabolites'. They regard methaemoglobin as problematic because it is considered an 'unvalidated biomarker of tafenoquine efficacy'.

The observation that tafenoquine and primaquine cause increases in blood methaemoglobin is uncontroversial. A recent analysis of different 8-aminoquinoline studies conducted over the past 80 years shows a clear relationship between radical curative efficacy and methaemoglobinaemia (*White et al., 2022*). There is also general agreement that the gametocytocidal and hypnozoitocidal activities of tafenoquine's predecessor primaquine result from the generation of reactive intermediates. The contentious part of the causal diagram in our paper is presumably that we consider tafenoquine to be similar, i.e. that the increase in methaemoglobin following tafenoquine is caused by its metabolites rather than the parent compound. Tafenoquine clinical trial data show that greater methaemoglobin values on day 7 are correlated with a lower probability of recurrence at 4 months (after adjustment for the mg/kg dose, *Watson et al., 2022*). This is the same relationship observed for primaquine and pamaquine (plasmoquine), suggesting these structurally related compounds all share a common mode of action. Importantly this hypothesis was prespecified before the tafenoquine meta-analysis (*White et al., 2022*). We believe that pre-registering the hypothesis that methaemoglobin is a biomarker of efficacy and then demonstrating an association in a large dataset provides strong evidence to support it as a biomarker.

Sharma et al. state: 'increases in blood MetHb concentrations after tafenoquine administration were highly correlated with mg/kg dose, but no correlation coefficients, indicating strength of correlation, were discussed in the manuscript'. Based on the responses to our first draft, there appears to be a mis-understanding of the word 'correlation'. Correlation denotes an association between two variables, not restricted to a causal relationship. Correlation can be quantified in a variety of ways,

not limited to covariance (defined as the expectation of the product of two random variables). We quantified the correlation between the tafenoquine mg/kg dose and the day 7 methaemoglobin using the linear model coefficient: 'Each additional mg/kg was associated with a 19% (95% CI: 17% to 21%) increase in day 7 MetHb concentrations'. The strength of the correlation is also made clear from Figure 3c.

## Potential safety concerns

The safety concern for tafenoquine is largely confined to patients with G6PD deficiency. There are two discrete sequential risk distributions for these G6PD deficient patients, neither of which have been well characterised. First, there is the probability of errors with quantitative testing in 'real world' use. Second, there is the extent of haemolysis in (a) hemizygote and homozygote patients and (b) heterozygotes with different G6PD deficiency variants. No G6PD deficient patient should receive tafenoquine with current quantitative testing. So ideally these dangerous events will not occur, but mistakes will happen. There are no available data on how frequently testing mistakes will occur. We also have no evidence to compare the subsequent haemolytic risks between the 5 mg/kg and 7.5 mg/kg doses.

## Discussion

Sharma et al. state that 'off-label use of a dose not robustly evaluated in clinical trials would pose a considerable risk to patient safety'. We have not recommended 'off-label' use of higher doses of tafenoquine. Our main recommendation was that clinical trials of higher doses should be conducted with the aim of providing optimal efficacy in patients at greatest risk of relapse.

Sharma et al. mention that 'single doses of tafenoquine 300 mg and 600 mg had similar relapse-free efficacy at 6 months (89.2% and 91.9%, respectively)' in the DETECTIVE Phase 2b study. As we noted above, this ignores variation in body weight and is based on a small sample size. Our pooled individual patient data meta-analysis suggests a substantial difference in efficacy between 5 and 10 mg/kg doses (*Watson et al., 2022*).

## Conclusion

The individual patient data pharmacometric meta-analysis of the tafenoquine pre-registration studies provides an evidence-based characterisation of the dose-response relationship for radical curative efficacy in vivax malaria. The results demonstrate clearly that when using the current 5 mg/kg regimen a substantial proportion of adults will be under-dosed, and therefore that there would be a substantial benefit from increasing the dose to 7.5 mg/kg. There is no need to wait for 'real-world evidence' to confirm this finding. The primary concern is safety: what would happen if a G6PD deficient patient was given tafenoquine radical cure by mistake? In this scenario the relative haemolytic risks for the 5 mg/kg and 7.5 mg/kg are not known. The development and regulatory approval of tafenoquine has provided a critical alternative for vivax radical cure in endemic countries. Our analysis supports the need for further prospective clinical trials of higher tafenoquine doses to characterise their efficacy, safety and tolerability.

## Additional information

### Competing interests

Justin A Green, Gavin CKW Koh: Former employee of GSK; shareholder in GSK. The other authors declare that no competing interests exist.

### Funding

| Funder | Grant reference number | Author |
|---|---|---|
| Wellcome Trust | 093956/Z/10/C | Nicholas J White |
| Wellcome Trust | 223253/Z/21/Z | James A Watson |

| Funder | Grant reference number | Author |
|---|---|---|

The funders had no role in study design, data collection and interpretation, or the decision to submit the work for publication. For the purpose of Open Access, the authors have applied a CC BY public copyright license to any Author Accepted Manuscript version arising from this submission.

## Author contributions

James A Watson, Conceptualization, Writing – original draft, Writing – review and editing; Robert J Commons, Joel Tarning, Julie A Simpson, Alejandro Llanos Cuentas, Marcus VG Lacerda, Justin A Green, Gavin CKW Koh, Cindy S Chu, François H Nosten, Richard N Price, Nicholas PJ Day, Writing – review and editing; Nicholas J White, Writing – original draft, Writing – review and editing

## Author ORCIDs

James A Watson ⓘ http://orcid.org/0000-0001-5524-0325
Robert J Commons ⓘ http://orcid.org/0000-0002-3359-5632
Joel Tarning ⓘ https://orcid.org/0000-0003-4566-4030
Julie A Simpson ⓘ http://orcid.org/0000-0002-2660-2013
Alejandro Llanos Cuentas ⓘ http://orcid.org/0000-0002-7567-5534
Marcus VG Lacerda ⓘ http://orcid.org/0000-0003-3374-9985
Gavin CKW Koh ⓘ https://orcid.org/0000-0002-7336-1566
Cindy S Chu ⓘ https://orcid.org/0000-0001-9465-8214
François H Nosten ⓘ https://orcid.org/0000-0002-7951-0745
Richard N Price ⓘ https://orcid.org/0000-0003-2000-2874
Nicholas PJ Day ⓘ https://orcid.org/0000-0003-2309-1171
Nicholas J White ⓘ https://orcid.org/0000-0002-1897-1978

## Ethics

Human subjects: Anonymised individual patient data were obtained via ClinicalStudyDataRequest.com following approval of a research proposal from the Independent Review Panel. Re-use of existing, appropriately anonymised, human data does not require ethical approval under the Oxford Tropical Research Ethics Committee regulations (OxTREC).

## Decision letter and Author response

Decision letter https://doi.org/10.7554/eLife.91283.sa1
Author response https://doi.org/10.7554/eLife.91283.sa2

# Additional files

## Supplementary files

• MDAR checklist

## Data availability

The data used in this study can be accessed by submitting a research proposal via https://www.clinicalstudydatarequest.com/ using the following study codes: GSK-TAF112582; GSK-200951; GSK-TAF116564. Decisions to share data with independent researchers are made via an Independent Review Panel. All code used to process the data is available on GitHub (copy archived at *Watson, 2023*).

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
