## [Decision Letter]

In the interests of transparency, eLife includes the editorial decision letter and accompanying author responses. A lightly edited version of the letter sent to the authors after peer review is shown, indicating the most substantive concerns; minor comments are not usually included.

Thank you for submitting your work entitled "Response to the comment on 'The clinical pharmacology of tafenoquine in the radical cure of *Plasmodium vivax* malaria: An individual patient data meta-analysis'" for further consideration by eLife. Your revised article has been evaluated by Dominique Soldati-Favre (as Senior Editor), Urszula Krzych (as Reviewing Editor), and a Reviewer (who has chosen to remain anonymous).

Please consider the comments and suggestions made by the Reviewer (please see below) and respond accordingly. We will also write to you separately about some editorial issues that need to be addressed.

*Reviewer #1:*

I have reviewed:

i) the comment from Sharma et al at GSK about the original eLife paper by Watson et al

ii) the response from Watson et al to the comment from Sharma et al.

I think both letters should be published substantially as they appear currently.

I do not agree with everything in Sharma et al regarding their defense of the original dosing regimen of 300mg Tafenoquine (plus chloroquine). In particular, Sharma et al shows an acute (hyper) sensitivity to safety issues which should be moot if the drug is truly given only to those who have been tested for G6PD deficiency. To base one's argument solely on those who are failing to meet this stated standard seems unusually strict especially when there are valid arguments for a higher dose for efficacy.

I also disagree with Watson's point "that tafenoquine is fundamentally a more dangerous drug". Yes, it is only given as a single dose which cannot be withdrawn, but tafenoquine's slow metabolism to the redox active form means that severe hemolytic events are very unlikely even if given inadvertently to G6PD deficient persons. There is a difference between grams of hemoglobin lost gradually over days and the more dangerous sudden hemolytic event which can cascade into a crisis. I would argue that primaquine is actually more likely to produce a dangerous event in those who hyper metabolize a single dose in a G6PD deficient person.

Arguments about efficacy are largely limited to SE Asia/Oceania and I find it strange that neither letter states that tropical Asian vivax will require more tafenoquine than other geographic areas since this is well known with its primaquine predecessor. Using the total drug data set which includes mostly Latin American studies to argue against increasing the drug regimen in SE Asia seems disingenuous. I really think the arguments should be focused on the rapidly and frequently relapsing malaria of SE Asia as that is the actual clinical problem. Sharma et al should not be diluting the issue with efficacy data mostly from Latin America as that is not where the problem is located.

Methaemoglobin (MetHb) may not be a validated endpoint but it is an important indicator of redox active metabolism. GSK's initial error with tafenoquine was to regard the drug as not metabolized on the basis of in vitro liver microsome studies. The INSPECTOR study [1] clearly indicates no efficacy against relapse when tafenoquine is combined not with chloroquine, but artemisinin. MetHb is the critical marker, likely of drug metabolism to a redox active intermediate. I think it is highly likely that INSPECTOR reflects lack of metabolism to the 5,6 orthoquinone and substantially reinforces Watson's argument that more studies are needed.

As such additional studies are evidently in process (a DSMB is being formed), it seems that both letters should reflect that more data is needed specifically focused on areas with frequently relapsing vivax malaria (SE Asia) in order to better define whether a higher dose regimen of tafenoquine is indicated or not.

[1] Sutanto I, et al. 2023 The Lancet Infectious Diseases. DOI: https://doi.org/10.1016/S1473-3099(23)00213-X

*Editorial issues to be addressed*:

Please find attached a revised version of the Scientific Correspondence article by Sharma et al. about your 2022 paper in eLife, along with a document containing a response from Sharma et al to a number of editorial queries from me.

Can you now please make the following revisions to your article.

# Please address the comments made by the reviewer.

# I asked Sharma et al to revise their article in response to the comments on lines 61-63, 86-89, 170-175, and 187-190 of your article. They have made some changes as a result of this request, so please revise your article accordingly.

# Sharma et al also made a number of comments on your article, I would be grateful if you could - as they request - revise lines 51-52 to avoid the term "more dangerous drug". You are free to revise your article in response to their other comments if you wish, but this is not essential.

# Please also make the following editorial changes

Abstract: Please reword as follows.

In our recent paper on the clinical pharmacology of tafenoquine (Watson et al., 2022) we used all available individual patient pharmacometric data from the tafenoquine pre-registration clinical efficacy trials to characterise the determinants of anti-relapse efficacy in tropical vivax malaria. We concluded that the currently recommended dose of tafenoquine (300 mg in adults, 5 mg/kg) is insufficient for cure in all adults, and that a 50% increase to 7.5 mg/kg would halve the risk of vivax recurrence by four months. We recommended that clinical trials of higher doses should be carried out to assess their safety and tolerability. Sharma et al. defend the currently recommended adult dose of 5 mg/kg as the optimum balance between radical curative efficacy and haemolytic toxicity (Sharma et al., 2023). We contend that the relative haemolytic risks of the 5 mg/kg and 7.5 mg/kg doses have not been sufficiently well characterised to justify this opinion. In contrast, we provided evidence that the currently recommended 5 mg/kg dose results in sub-maximal efficacy, and that prospective clinical trials of higher doses are warranted to assess their risks and benefits.

---

## [Author Response]

Reviewer comments:

We thank the reviewer for reading our reply in detail.

“I also disagree with Watson’s point “that tafenoquine is fundamentally a more dangerous drug”. Yes, it is only given as a single dose which cannot be withdrawn, but tafenoquine’s slow metabolism to the redox active form means that severe hemolytic events are very unlikely even if given inadvertently to G6PD deficient persons. There is a difference between grams of hemoglobin lost gradually over days and the more dangerous sudden hemolytic event which can cascade into a crisis. I would argue that primaquine is actually more likely to produce a dangerous event in those who hyper metabolize a single dose in a G6PD deficient person.”

We agree that this phrasing was too strong. Although 8-aminoquinoline induced haemolysis is inevitable in G6PD deficiency, the extent of haemolysis is highly variable. We do not know how a G6PD hemi or homozygote would response to a high tafenoquine dose. Although we agree the metabolism is slower, this remains an open question and as such tafenoquine should be considered highly dangerous in G6PD deficiency.

Editorial comments:

We have made all the requested changes, apart from two points:

• In the abstract we have kept “Sharma et al, representing the pharmaceutical company GSK”: it is important to point out that this criticism has a clear conflict of interest. This point is especially relevant now that the authors from MMV have removed their names from the letter.

• We have changed “tafenoquine is fundamentally a more dangerous drug” to “Compared with primaquine, the most widely used 8-aminoquinoline antimalarial, tafenoquine, if not used appropriately, may have a higher risk of causing severe haemolysis.”. See reply to reviewer above.